



# Data reduction of incoherent scatter plasma line parameters

Mini Gupta[1] and Patrick Guio[1,2]

[1]Department of Physics & Technology, UiT The Arctic University of Norway, Tromsø, Norway
[2]Department of Physics and Astronomy, University College London, London, UK

**Correspondence:** Mini Gupta (mini.gupta@uit.no)

**Abstract.** In the ionosphere, a sustained population of suprathermal electrons arises due to photoionization or electron precipitation. The presence of such a population enhances the scattered power in the plasma line spectrum, thus making it possible to detect them. Plasma line measurements improve the accuracy of electron density and temperature estimates. We investigate plasma line enhancements in EISCAT Tromsø UHF radar observations, using two image processing methodologies for detection: a supervised image morphological processing technique and an unsupervised connected component analysis. The supervised methodology detects more plasma lines, demonstrating higher sensitivity. We determine the times and altitudes with enhancements and model the spectrum with a Gaussian function. The radar beam points in the field-aligned direction for 25% of the total observational time, is directed east for another 25% and is oriented in the vertical direction for the remaining 50%. Plasma lines are detected 26% of the time when the radar is pointed in the field-aligned direction, 5% of the time in the east direction and 5% of the time in the vertical direction. Most plasma lines are detected around the F-region altitude where the electron density is maximum, typically between 230–260 km, with a simultaneous increase in the electron density estimates from the ion line. Plasma line intensity is maximum around noon. It decreases as the aspect angle increases. Both detection methodologies' advantages and disadvantages are discussed, and plasma line intensity variations are analyzed as a function of altitude, aspect angle and phase energy.

## 1 Introduction

Powerful electromagnetic pulses are transmitted by ISRs into the ionosphere, causing electrons to oscillate and re-radiate along the radar scattering wave vector. The scattering wave vector depends on the radar operating frequency and geometry. The scattered signal is collected by the radar and analyzed as a power spectrum (Dougherty and Farley, 1960). ISRs probe the ionospheric plasma at a wavelength much greater than the Debye length, $\lambda_D$ i.e.

$$\alpha = \frac{1}{k_s \lambda_D} \gg 1, \tag{1}$$

where $k_s = 2\pi/\lambda_s$, $k_s$ and $\lambda_s$ are the scattering wavenumber and wavelength, respectively. The electron Debye length is defined as $\lambda_D = \sqrt{\epsilon_0 k_B T_e/n_e e^2}$ where $\epsilon_0$ is the permittivity of free space, $k_B$ is the Boltzmann constant, $T_e$ is the electron temperature, $n_e$ is the electron density and $e$ is the electron charge. ISRs provide valuable information about the collective interactions in the plasma. Most of the scattered power is present at pairs of frequencies corresponding to the natural electrostatic modes of the plasma, where the resonance condition of the magnetized plasmas is satisfied (Evans, 1969). The Doppler shift of each





frequency can be either positive, indicating a wave travelling towards the radar, or negative, signifying a wave travelling away along the scattering direction. These resonant frequency pairs are classified as ion lines and plasma lines, corresponding to ion acoustic waves and Langmuir waves, respectively (Akbari et al., 2017).

Most of the scattered power is contained within the ion lines. The integrated power of the ion line in thermal equilibrium is given by (Bauer, 1975; Fredriksen et al., 1992)

$$I_i = \frac{\alpha^4}{(1 + \alpha^2)(1 + \alpha^2 + \alpha^2 T_e/T_i)}, \tag{2}$$

where $T_i$ is the ion temperature. The ion line spectrum is fitted to a model power spectrum, under the assumption of scatter from a uniform plasma in thermal equilibrium i.e. the electron and ion velocity distributions are Maxwellian. The three-dimensional isotropic Maxwellian velocity distribution is given by

$$f(\boldsymbol{v}) = \left(\frac{m_q}{2\pi k_B T_q}\right)^{\frac{3}{2}} \exp\left(-\frac{m_q(\boldsymbol{v} - \boldsymbol{u}_q)^2}{2 k_B T_q}\right), \tag{3}$$

where $m_q$ is the mass, $T_q$ is the temperature, and $\boldsymbol{u}_q$ is the bulk drift velocity of species $q$ (either electron $e$ or ion $i$). The ionospheric plasma parameters $n_e, T_e, T_i$ and $u_i$ are estimated from the ion line for each range gate. In this case, the range gate

is equal to the spatial resolution of the analysis. These parameters characterize the uniform background ionospheric plasma along the direction of the scattering wavevector (Kudeki and Milla, 2011). For a thermal plasma, the integrated power in one plasma line is written as:

$$I_p = \frac{1}{2\alpha^2}. \tag{4}$$

When the condition Eq. (1) is satisfied, then the ratio of the integrated power in the plasma line ($I_p$) and the ion line ($I_i$) is given by (Bauer, 1975)

$$\frac{I_p}{I_i} \approx \frac{1}{2\alpha^2}\left(1 + \frac{T_e}{T_i}\right). \tag{5}$$

This implies that plasma lines generally contain less power than the ion lines, as seen in Eq. (5), and are difficult to observe in uniform plasmas in thermal equilibrium. However, photoionization or auroral precipitation can generate a suprathermal population and the power in the plasma line spectra can be enhanced (Perkins and Salpeter, 1965).

Plasma line measurement enables the study of several properties that are not directly available with only the measurement of the ion lines. For example, plasma lines allow for the analysis of fine structures in suprathermal electron velocity distribution

at velocities imposed by the plasma line frequency and the radar frequency (Guio and Lilensten, 1999). Electron density and temperature estimates are also improved by plasma lines (Nicolls et al., 2006). Plasma line asymmetry can be used to measure electron drift velocity and electron density, which, together with ion drift velocity, provide an estimate of the ionospheric current (Guio et al., 1996). Plasma line measurements facilitate the determination of additional parameters, like ion composition (Bjørnå and Kirkwood, 1988; Fredriksen et al., 1989) and ion-neutral collision frequency (Bjørnå, 1989) by providing extra

constraints, reducing ambiguity between parameters.





Plasma lines have been the subject of numerous studies. Investigations of plasma lines along the magnetic field at high latitudes have encompassed experimental studies (Guio et al., 1996), theoretical modelling (Nilsson et al., 1996) as well as comparing experimental data with theoretical modelling (Guio and Lilensten, 1999). Plasma line parameters at oblique angles to the magnetic field have also been explored through experimental measurements at low latitudes (Djuth et al., 2018) and
comparative studies combining theoretical models with experimental data at both low latitudes (Fremouw et al., 1969; Longley et al., 2021) and high latitudes (Fredriksen et al., 1992; Kirkwood et al., 1995). Kirkwood et al. (1995) focused exclusively on plasma lines in the E-region, while Fredriksen et al. (1992) considered plasma lines in the F-region as well. However, these studies do not address high-latitude F-region plasma line intensity measurements as a function of phase energy at low aspect angles, a gap we aim to fill.

In Section 2 we present a description of the EISCAT UHF radar experimental setup and IP2 scanning technique used to collect the data presented here. Section 3 covers the general methodologies for supervised and unsupervised detection of plasma lines, parameter extraction using Gaussian model fitting and intensity derivation. In Section 4, we summarize the main findings from applying these methodologies to ISR data collected with the EISCAT UHF radar in Tromsø between 07:00–15:00 UT from 26 to 31 January 2022. Finally, Section 5 discusses the advantages and disadvantages of the supervised and unsupervised
detection methodologies, examines plasma line intensity variations as a function of time, altitude, phase energy, and the aspect angle between the magnetic field and the scattering wavevector, and suggests areas for future research.

## 2 Experiment

The data analyzed in this study was collected with the EISCAT UHF radar in a scanning sequence as part of a common programme experiment known as the IP2 or CP2 experiment. The IP2 scan consists of a beam-swing sequence where the
antenna points in three directions in a short cycle. The details of the radar scan sequence are described in Table 1. It takes $15\,\mathrm{s}$ for the radar to manoeuvre between pointing directions, and therefore, the radar scan cycle duration is $4\,\mathrm{min}$. The scattering wavevector is the difference between the received and transmitted wavevector i.e. $\boldsymbol{k}_s = \boldsymbol{k}_r - \boldsymbol{k}_t$. The radar is being used in monostatic mode i.e. the same antenna to transmit and receive. Therefore, the scattering wavevector when receiving at the radar operating frequency is $\boldsymbol{k}_s = -2\boldsymbol{k}_t$ with wavenumber $k_s = 4\pi f_{\mathrm{radar}}/c$, where $f_{\mathrm{radar}}$ is the frequency of the operating radar
and $c$ is the speed of light in vacuum. For the EISCAT UHF radar, $k_s = 39\,\mathrm{m}^{-1}$.

The radar has a transmitter frequency of $927\,\mathrm{MHz}$ (wavelength $\lambda = 32.2\,\mathrm{cm}$), a peak power of $1\,\mathrm{MW}$ and a $11\,\%$ duty cycle. The experiment uses a 32-bit alternating code transmitter scheme with a baud length of $20\,\mu\mathrm{s}$. Each code consists of 64 sub-cycles of duration $5.58\,\mathrm{ms}$, resulting in a total cycle duration of $0.357\,\mathrm{s}$. Measurements have a pre-integration time of $5\,\mathrm{s}$, with a total of $14$ cycles included in each data dump. It is assumed that the ionosphere remains stationary throughout this time
resolution.

The raw data collected by the radar consists of the autocorrelation function (ACF) evaluated at discrete time lags (Lehtinen and Häggström, 1987). Plasma line ACFs are sampled with a lag increment of $0.4\,\mu\mathrm{s}$, and the maximum lag is $320\,\mu\mathrm{s}$, resulting in a total of 800 lags. The power spectrum is derived by taking the Fourier transform of the ACF. The resulting frequency





**Table 1.** Radar scan direction and angles

| Pointing Direction | Elevation Angle | Azimuth Angle | Dwell time | Number of data dumps | Aspect angle at 240 km |
|---|---|---|---|---|---|
| Field-Aligned (at 240 km) | 77.8° | 189° | 60 s | 12 | 0° |
| Vertical | 90° | 261.1° | 30 s | 6 | 12° |
| East | 75.4° | 259.6° | 60 s | 12 | 15° |
| Vertical | 90° | 190.5° | 30 s | 6 | 12° |

resolution of the power spectrum is 1.56 kHz and a bandwidth of 2.5 MHz. The spectra consist mostly of white noise, with the possibility of a narrow-banded plasma line signal being present. Plasma line data from two downshifted receiver bands (between −5.25 and −2.75 MHz, and between −7.65 and −5.15 MHz) were analyzed. Data were examined from 07:00 UT to 15:00 UT every day between 26–31 January 2022. The spectral and range characteristics of the ion line and plasma line channels are given in Table 2.

**Table 2.** Characteristics of Plasma and Ion Line Channels

| Channel Type | Spectral Resolution | Bandwidth | Range Span | Range Resolution |
|---|---|---|---|---|
| Plasma Line | 1.56 kHz | 2.5 MHz | 107–374 km | 3 km |
| Ion Line | 1.22 kHz | 100 kHz | 49–693 km | 1.5 km |

# 3 Methodology

## 3.1 Derivation of Plasma Line Spectra

We used GUISDAP (Lehtinen and Huuskonen, 1996) for analyzing ion lines and extended its capabilities to assess plasma lines. The analysis was performed on the collected data from the EISCAT UHF radar. The data was integrated over each position's dwell time, followed by background subtraction and calibration. The power spectra were obtained by taking a Fourier transform of the ACF. At this stage, we have, for each time, two-dimensional data depicting the plasma line antenna temperature as a function of frequency and altitude as depicted in the upper left plots in Figures 1 and 2.

## 3.2 Plasma Line Detection

To identify times and altitudes with enhanced plasma lines, we implemented two methodologies: a supervised one based on image morphological processing and an unsupervised one based on connected components. Both methodologies begin with the same initial steps: median filtering of the spectra and noise removal, as described by Ivchenko et al. (2017). A median filter





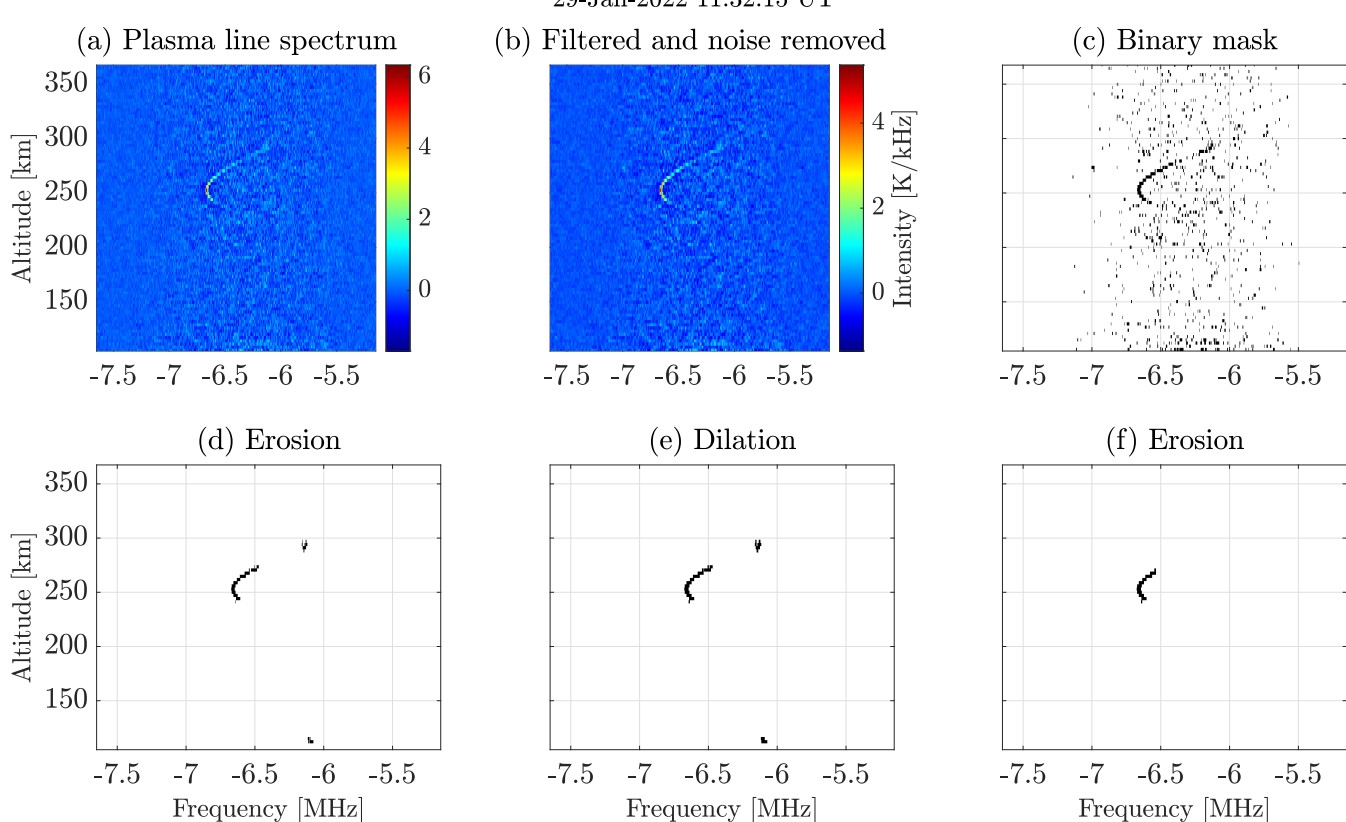

**Figure 1.** Image morphological processing steps for plasma line detection: (a) plasma line spectra as a function of altitude and frequency with $60\,\mathrm{s}$ integration time and $0°$ aspect angle, (b) median filtering (3 point frequency width) and background noise removal, (c) binary mask creation (threshold at $3.98\times$MAD), (d) removal of regions with less than 16 connected points, (e) convolution with the rectangular kernel (1 range gate $\times$ 3 frequency bins), and (f) removal of regions with less than 47 connected points.

of size $1\times3$ was applied, where 1 and 3 are the number of points in the range and frequency dimensions, respectively. Noise and clutter were removed using the methodology described in Ivchenko et al. (2017). The spectra obtained after filtering and removing noise are shown in the upper middle plot of Figure 1 and the upper right plot of Figure 2.

Additionally, both methodologies use the median absolute deviation estimator, denoted MAD, to assess the dispersion in the spectra and set thresholds for detecting points with high power in the spectral image. Given a sample dataset, $x$, with a median

value of $\tilde{x} = \mathrm{median}(x)$, the MAD estimator is defined as (Wilrich, 2007):

$$\mathrm{MAD} = \mathrm{median}(|x - \tilde{x}|). \tag{6}$$

For the examples shown in Figures 1 and 2, the MAD for the filtered and noise-removed image is calculated to be $1.2\times10^{-4}\,\mathrm{K/kHz}$. A discussion of both methodologies follows below.




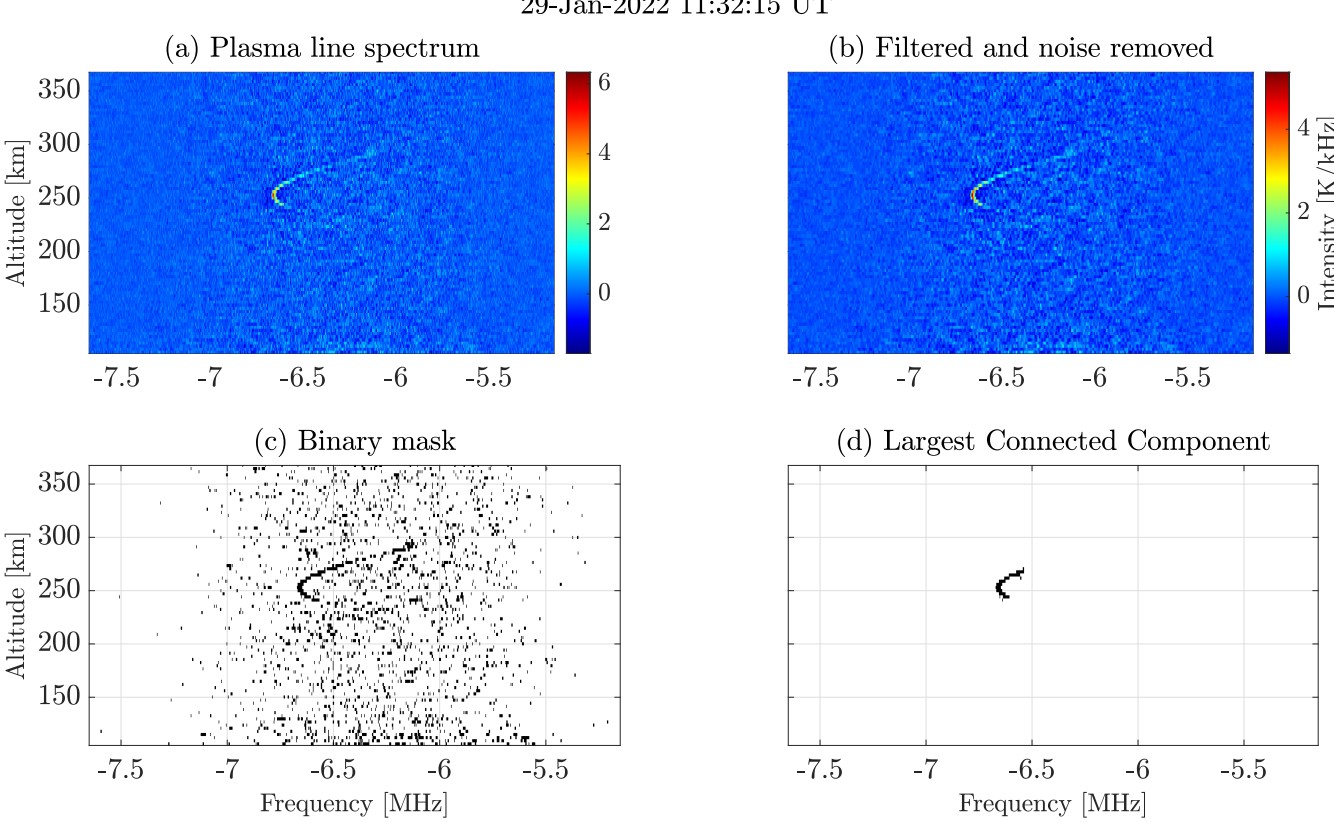

**Figure 2.** Plasma line detection using connected component analysis: (a) and (b) are the same as in Figure 1, (c) threshold $3.4 \times 10^{-3}$ K/kHz on (b) to convert to binary image, (d) largest connected component identifying the plasma line.

### 3.2.1 Supervised Methodology: Image Morphological processing

We implemented the technique proposed by Ivchenko et al. (2017) but with different parameters for thresholding and image
morphological processing tailored to each channel for detecting enhanced profile points. We calibrated the image morphological processing parameters through iterative manual adjustment, examining the spectra to avoid false positives.

For the first downshifted channel ($-5.25$ to $-2.75$ MHz), a threshold of $3.95 \times$ MAD was set to obtain a binary mask. Regions with fewer than 10 connected points were then eroded. This was followed by dilation using a structuring element sized $1 \times 3$, where 1 corresponds to the range (no dilation) and the frequency (dilation across 3 points). Subsequently, regions with
less than 45 connected points were removed to yield the final mask containing the plasma line signal.

In the case of the second downshifted channel ($-7.65$ to $-5.15$ MHz), a threshold of $3.98 \times$ MAD was applied. This was followed by the erosion of areas containing fewer than 16 connected points, dilation using the same $1 \times 3$ structuring element as in the first downshifted channel, and a subsequent erosion targeting areas with less than 47 connected points.





In our study, the values for binary thresholding, erosion, and dilation are smaller in comparison to those used by Ivchenko
et al. (2017). This difference is due to the signal-to-noise ratios (SNR) with EISCAT UHF measurements having higher SNR
than the EISCAT Svalbard radar, as highlighted by Nilsson et al. (1996).

### 3.2.2 Unsupervised Methodology: Connected Component Analysis

MAD approximates the width $w$ of the $50\%$-interval around the median $\tilde{x}$ of the distribution of $x$ as $P(|x-\tilde{x}|\leq w)=0.5$.
Under the assumption of normality in the data distribution, $w$ can be expressed as:

$$w = z_{0.75}\sigma = 0.6745\sigma, \tag{7}$$

with $z_{0.75}$ being the $75\%$-quantile of the standard normal distribution and $\sigma$ is the standard deviation. However, this estimator
is inherently biased and tends to underestimate variability. To obtain an unbiased estimate under normality, we apply a bias
correction to the MAD estimator. The bias-corrected median absolution deviation is (Daszykowski et al., 2007)

$$\sigma_{\mathrm{MAD}} = \mathrm{MAD}/0.6745 = 1.4826 \cdot \mathrm{MAD}. \tag{8}$$

This correction gives an unbiased estimator with an asymptotic efficiency of $36.7\%$ and an asymptotic breakdown point of
$50\%$, meaning it remains reliable even when up to $50\%$ of the data are outliers.

In this analysis, we convert the spectral images into binary images by applying a threshold. A fixed threshold of $2\sigma_{\mathrm{MAD}}$ is set
to identify pixels with high intensity. After thresholding, each binary image contains multiple connected components, which
are groups of adjacent pixels with intensity higher than the threshold. An example is seen in the lower left plot of Figure 2,
where black regions represent distinct connected components. The size of each connected component is given by the number
of pixels it comprises. To identify the plasma line signal, we focus on the largest connected component in each binary image,
based on the expectation that a significant enhancement in the plasma line signal will result in the largest connected component.

If $A_i$ denotes the number of pixels in the largest connected component of the $i^{\mathrm{th}}$ image and there are a total of $N$ spectral
images, then the mean, $\mu$, and standard deviation, $\sigma$ of these largest components across all images are given as:

$$\mu = \frac{1}{N} \sum_{i=1}^{N} A_i, \tag{9}$$

$$\sigma = \sqrt{\frac{1}{N-1} \sum_{i=1}^{N} |A_i - \mu|^2}. \tag{10}$$

Let $A_{\min}$ represent the minimum number of connected pixels that the largest component in a binary image must contain to be
classified as a plasma line signal. The threshold $A_{\min}$ for detecting plasma lines is set at the three-sigma detection limit, defined
as $A_{\min}=\mu+3\sigma$. The rationale for this threshold is that over a full day of high-frequency spectra recordings, most spectra will
contain only random noise. Consequently, the size of the largest connected component in these noisy images will typically be
small. Plasma line signals are detectable only during rare specific events such as strong photoionization or auroral precipitation.



**Figure 3.** Histogram of pixel counts in the largest connected component from each image on 29 January 2022. The blue line is the mean, $\mu = 30$, while the green line is the threshold, $A_{\min} = \mu + 3\sigma = 93$, on the size of the largest connected component for plasma line detection.





During these events, the plasma line signal results in a much larger size of the largest connected component. Statistically, this
means that events containing plasma lines are outliers. An example is shown in Figure 3, where a histogram of the largest
connected component sizes is plotted for an entire day of spectral recordings. A peak at a size of 30 pixels represents noise-
dominated images (Figure 3). As the size of the largest connected component increases, the histogram decreases, reflecting the
rarity of events with the larger size of the largest connected components. The mean size is $\mu{=}30$, with a standard deviation
of $\sigma{=}21$, yielding a threshold of $A_{\min}{=}93$ pixels (denoted by the green line in Figure 3). Any spectral image with the largest
connected component of 93 pixels or more is detected as containing an enhanced plasma line, as shown in Figure 2. Although
this threshold $A_{\min}$ may seem high, it effectively avoids false positives in plasma line detection.

$A_{\min}$ is calculated separately for each day and each channel. It is determined automatically from the data as described above
rather than manually tuned, ensuring the methodology remains unsupervised. The identification of plasma lines is based only
on the statistical outlier of the size of the largest connected components, which makes this methodology data-driven, and
hence, unsupervised. This approach is necessary because the noise levels can vary between channels, and calculating a single
threshold across all channels could skew the result due to these differences in noise distribution.

### 3.3   Plasma Line Parameter Extraction

At this stage, we have identified the detection times of the plasma lines. The altitudes where plasma line enhancement occurs
are indicated by the final mask in the bottom-right plots of Figures 1 and 2. At these altitudes, the plasma line is modelled
using a Gaussian function characterized by three parameters: intensity $A_p$, frequency $f_r$, and half-bandwidth $\delta_f$. To extract
these parameters, we used a non-linear, unweighted least-squares Levenberg–Marquardt algorithm, similar to the approach
proposed by Guio et al. (1996). As shown in Figure 4, the spectra were fitted to a Gaussian function

$$S(f; A_p, f_r, \delta_f) = \frac{A_p}{\sqrt{2\pi}\delta_f} \exp\left(-\frac{(f - f_r)^2}{2\delta_f^2}\right). \tag{11}$$

Figure 5 shows altitude profiles of the extracted parameters from Figure 4 and their uncertainties. The uncertainties were
estimated for a $95\,\%$ confidence intervals around the parameter estimates.

### 3.4   Derivation of Plasma Line Temperature

A more meaningful form of expressing plasma line intensity is in terms of plasma line temperature (Yngvesson and Perkins,
1968; Fredriksen et al., 1992; Guio et al., 1996; Nilsson et al., 1996) because it allows comparing the plasma line intensity
with the theoretical prediction (Fredriksen et al., 1992; Guio et al., 1998) and helps understand how much the plasma line is
enhanced above the thermal level. Simultaneous ion and plasma line measurements can be utilized to derive the plasma line
temperature (Yngvesson and Perkins, 1968; Fredriksen et al., 1992). In the case of a monostatic radar, the total received power
in the plasma line can be expressed by the following radar equation (Yngvesson and Perkins, 1968)

$$k_B T_A^p B_W = \frac{P_T}{R^2} r_e^2 \sigma_P \frac{c\tau}{2} A(\nu), \tag{12}$$





**Figure 4.** Plasma line spectra from Figure 1 (a) (top left plot) at altitudes selected by reduced mask in Figure 1 (f) (bottom right plot). The black lines are ISR spectra at selected altitudes, while the red lines are Gaussian model fit.



**Figure 5.** Altitude profiles of the plasma line parameters (intensity $A_p$, frequency $f_r$ and half-bandwidth $\delta_f$) from the fit in Figure 4. Error bars show 95 % confidence level in the fitted parameters.

where $T_A^p$ is the antenna temperature of the plasma line, $B_W$ is the bandwidth of the plasma line channel, $P_T$ is the transmitted power, $R$ is the distance between the radar and the volume contributing to the plasma line intensity, $r_e$ is the classical electron radius, $\frac{c\tau}{2}$ is the length of the range-gate that contributes to the plasma line signal (assuming uniform power distribution over the volume), and $A(\nu)$ is the effective antenna area at frequency $\nu$. $\sigma_P$ is the scattering cross-section contributing to the plasma line, given by (Yngvesson and Perkins, 1968; Guio, 1998)

$$\sigma_P = \frac{T_p}{T_e} \frac{1}{2} n_e k_s^2 \lambda_D^2, \tag{13}$$

where $T_p$ is the plasma line temperature, $k_s = 2\pi/\lambda$ is the scattering wavenumber where $\lambda = c/(2 f_{\text{radar}} + f_{\text{offset}})$ is the scattering wavelength of the radar operating at frequency $f_{\text{radar}}$ and $f_{\text{offset}}$ is the frequency offset at which plasma line peak is observed.





As seen in Eq. (13), the plasma line temperature depends on the scattering wavenumber. This means that ISRs operating at different frequencies have different sensitivities to plasma lines and probe different energy regions within the suprathermal electron velocity distribution. The resonance condition is satisfied when the velocity of the electrons matches the phase velocity of the radar scattering wave. This phase velocity is given by:

$$v_\phi = f_{\text{offset}}\lambda = \frac{f_{\text{offset}}c}{2f_{\text{radar}} + f_{\text{offset}}}. \tag{14}$$

Since the radar operating frequency is typically much higher than the plasma line frequency ($f_{\text{radar}} \gg f_{\text{offset}}$), the phase velocity

can be approximated as:

$$v_\phi \simeq \frac{f_{\text{offset}}c}{2f_{\text{radar}}}. \tag{15}$$

Therefore, the corresponding kinetic energy of the electrons is

$$E_\phi = \frac{1}{2}m_e v_\phi^2 = \frac{m_e f_{\text{offset}}^2 c^2}{8 f_{\text{radar}}^2}. \tag{16}$$

Thus, an ISR operating at a lower frequency (e.g., EISCAT VHF at $224\,\text{MHz}$) is more sensitive to electrons with larger velocities in the tail of the suprathermal velocity distribution. As a result, such radars can detect subtle features in the high-energy tail of the distribution like peaks at $24.25\,\text{eV}$ and $26.25\,\text{eV}$ caused by the ionization of $N_2$ and O from solar HeII

radiation (Guio and Lilensten, 1999) or auroral beams. Conversely, an ISR operating at a higher frequency (e.g., EISCAT UHF at $931\,\text{MHz}$) is more sensitive to electrons with lower velocities. Thus, a higher-frequency ISR might be better suited for studying features like the 2–4 eV $N_2$ dip (Fredriksen et al., 1992; Kirkwood et al., 1995; Nilsson et al., 1996). Using multiple ISRs at different frequencies gives the ability to probe different velocity ranges, making it possible to construct a picture of the plasma line spectra due to different features in the suprathermal velocity distribution.

## 4  Results

The analysis of plasma line data using the supervised methodology revealed detectable enhancements for about $10\%$ of the total observational time. Plasma lines were observed $26\%$, $5\%$ and $5\%$ of the total pointing time in the field-aligned, east and vertical directions, respectively. Over the entire analysis period (26-31 January 2022 with 8 hours per day between 07:00-15:00 UT, except on 26 January between 08:00-15:00 UT), the unsupervised methodology detected 110 plasma lines, while

the supervised methodology detected 275. Here, we present the results of the analysis for 29 January 2022 07:00-15:00 UT only.

The plasma line data fit well to the Gaussian model (see Figure 4), as was also shown in Guio et al. (1996). Figure 5 shows the altitude profile of the plasma line parameters derived from the fit in Figure 4. It can be seen that the derived plasma line frequency exhibits a smooth variation in altitude, with the maximum in the magnitude of plasma line frequency at the F2 peak

altitude of $251.5\,\text{km}$. This behaviour is due to the plasma frequency's quasi-parabolic variation with altitude around the peak at hmF2, as pointed out in Guio et al. (1996). Near the F-layer peak, the reduced frequency spreading leads to a stronger and





narrower plasma line at the altitude with the highest electron density. In contrast, the steeper gradient of the plasma frequency away from the peak causes more frequency spreading of the plasma line signals above or below the F-layer peak. This trend is evident in Figures 4 and 5, where the plasma line width is narrowest at the F2 peak and gradually broadens above and below this altitude.

Figure 6 shows the electron density and temperature estimates as a function of time and altitude obtained from GUISDAP using the ion line data from 29 January 2022. A scaling parameter known as the Magic constant must be specified in GUISDAP for the ion line analysis. The Magic constant is calibrated to ensure that the electron density derived from the plasma line matches that obtained from the ion line. A different Magic constant was calculated for each day of the experiment. GUISDAP assumes Maxwellian electron and ion velocity distributions to estimate these parameters. The parameters are plotted over the same time period, from $07{:}00\,\mathrm{UT}$ to $15{:}00\,\mathrm{UT}$, and the same altitude range, $107–374\,\mathrm{km}$ (see Table 2), where the plasma lines were analysed. The black vertical lines indicate the time interval during which plasma lines were detected using the supervised methodology as demonstrated in Figure 1. During this interval, electron density increases to $\approx 5\times 10^{11}\,\mathrm{m}^{-3}$, corresponding to a plasma frequency of $\approx 6\,\mathrm{MHz}$, between $230$ and $280\,\mathrm{km}$, i.e. around the F-layer peak where scale height is largest. As a result, most of the plasma lines observed are in the same altitude range. Figure 7 shows the estimated plasma line parameters as function of time and altitude derived using the plasma line parameter extraction technique (see example provided in Figures 4 and 5) from all positive plasma line detections at different times and altitudes on 29 January 2022, following the supervised methodology illustrated in Figure 1, with analysis similar to Guio et al. (1996).

The plasma line is observed only in the downshifted channel, with frequency magnitudes between $5.5$ to $6.7\,\mathrm{MHz}$. The plasma line frequency increases around $11{:}20{:}15\,\mathrm{UT}$ (Figure 7), driven by an increase in the photoelectron population from solar EUV radiation near noon. From Figure 7, it can be seen that the plasma line frequency gradually increases, following the sun exposure, from $10{:}00–11{:}20\,\mathrm{UT}$.

We have similar plots to those shown in Figures 6 and 7 for the other days. The background electron density and temperatures, as shown in Figure 6, do not vary significantly from day to day. The variation in background electron density over different days is between $\approx 2\times 10^{11}$ to $9\times 10^{11}\,\mathrm{m}^{-3}$. As can be seen from Eq. (16), the phase energy depends on the plasma line frequency and therefore the background electron density. Since the electron density naturally fluctuates from one day to another, the range of the phase energy explored will consequently vary.

The plasma line frequency bands extend from $-7.65$ to $-2.75\,\mathrm{MHz}$, as described in Section 2. Using Eq. (16), it can be determined that the EISCAT UHF system can probe an energy interval of $0.6–4.4\,\mathrm{eV}$. However, as shown in Figure 10, the observed plasma lines are confined to the energy range of $1.7–3.4\,\mathrm{eV}$. This indicates that, despite detecting hundreds of plasma lines, only $45\,\%$ of the total phase energy range accessible is effectively explored.

## 5 Conclusions

We discussed two methodologies for detecting plasma lines. The first, a supervised approach based on image morphological processing, requires manual tuning of binary threshold and erosion/dilation parameters. This approach requires an initial guess





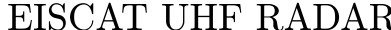

**Figure 6.** Electron density and temperature estimated from ion line analysis using GUISDAP. Black vertical lines indicate the period of plasma line detection by the supervised methodology, with electron density enhancements from photoionization observed between these lines.



**Figure 7.** Estimated parameters of the downshifted plasma line as a function of altitude and time on 29 January 2022, using the methodology illustrated in Figure 4 during the period marked by black vertical lines as shown in Figure 6. Top: Doppler frequency ($f_r$); middle: half-bandwidth ($\delta_f$); bottom: intensity ($A_p$). Colour bars indicate parameter magnitudes.



of the parameters followed by iterative adjustments. In our case, we began with the parameters used by Ivchenko et al. (2017) for ESR IPY experiments and refined them until no false positives were detected for a given day and channel. However, selecting parameters in this supervised approach relies heavily on trial and error, requiring many adjustments based on the observation day, radar channel, and radar type. In contrast, the methodology based on connected component analysis is an unsupervised approach, as the detection thresholds are not manually defined but are automatically derived from statistical estimators and the
distribution of the connected component sizes. This data-driven approach establishes thresholds for converting spectral images to binary format and for detecting plasma lines in entire datasets for a given day, channel, and radar, eliminating the need for initial assumptions or manual adjustments. As a result, the unsupervised approach is more adaptable to different radars.

The supervised methodology detected more plasma lines than the unsupervised. Both supervised and unsupervised methodologies were tuned to avoid false positives. The supervised methodology may have a higher sensitivity to weaker plasma lines,
while the unsupervised with statistically derived thresholds, is more conservative and may miss some weaker signals. Each methodology has its strengths depending on the goals of the analysis. The supervised methodology may be preferable when the goal is to maximize detection sensitivity, while the unsupervised methodology is more suitable for working with large datasets spanning multiple days or different radars.

Both methodologies are robust against noise factors such as (a) localized noise at specific altitudes, as shown in Figure 8,
and (b) higher noise around the centre frequency of the filters, as shown in Figure 9. The filtering and noise removal technique from Ivchenko et al. (2017) enhances plasma lines despite these challenges, facilitating plasma line profile detection.

$88\%$ of the plasma lines detected in the downshifted channels occurred at the magnitude of frequencies above $5.25\,\mathrm{MHz}$, with the lowest detected plasma line frequency across all six days being $4.7\,\mathrm{MHz}$. This means that the plasma lines detected in the downshifted channel between $2.75$ to $5.25\,\mathrm{MHz}$ occurred near the edge of the filter. In the experiment, no data were
recorded in the upshifted channel at frequencies above $5.25\,\mathrm{MHz}$. Although data were recorded at upshifted frequencies between $2.75$ to $5.25\,\mathrm{MHz}$, no plasma lines were detected in this range, possibly because plasma lines in downshifted channels likely occur above $5.25\,\mathrm{MHz}$. The absence of upshifted plasma lines means that we cannot use our data to calculate ionospheric currents.

We can nevertheless analyze plasma line intensity as a function of phase energy. This provides fine-grained details about
the suprathermal electron velocity distribution at the corresponding velocity range. The phase energy of the plasma line corresponds to the kinetic energy of the electrons that are in resonance with the electrostatic wave. Since the F10.7 index, the electron density and temperature do not exhibit substantial variations across the entire analysis period, we combine data from each day (26 to 31 January 2022 07-15 UT) to plot the plasma line intensity, $k_B T_p$, as a function of phase energy, $E_\phi$, calculated from the derived plasma line frequency using Eq. (16).

Figure 10 shows plasma line intensity versus phase energy at different altitudes, using plasma line measurements from 26 to 31 January 2022. Plasma line intensity is a function of time, altitude, phase energy and aspect angle; here, we integrate it over time and aspect angles to display it as a function of phase energy and altitude. The figure uses box plots to represent plasma line intensity across uniform phase energy intervals: each box's central line shows the median plasma line intensity, with top and bottom edges indicating the upper and lower quartiles (the 75th and 25th percentiles, respectively). The interquartile range



30-Jan-2022 09:20:15 UT

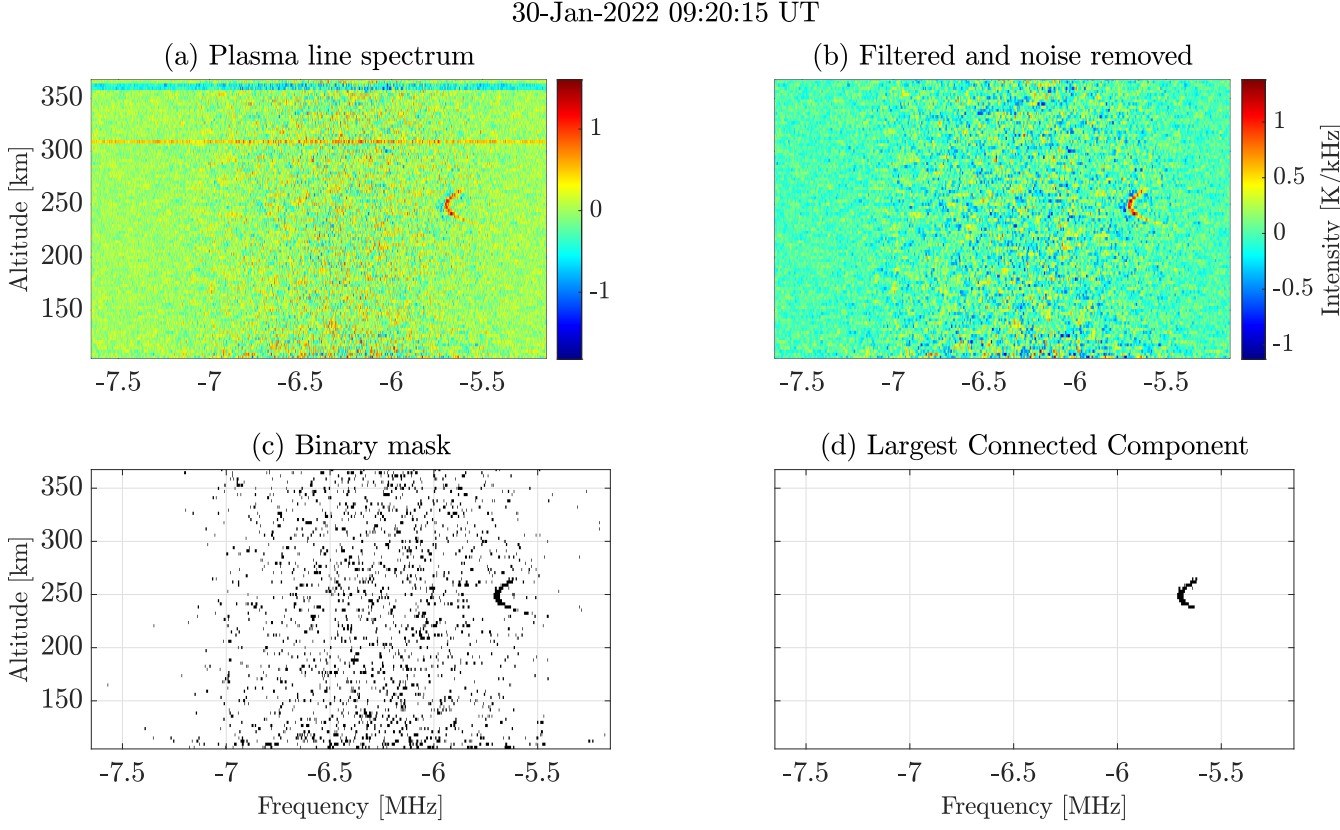

**Figure 8.** (a) Plasma line spectrum recorded by EISCAT UHF radar, showing strong localized noise at $307\,\mathrm{km}$ and $360\,\mathrm{km}$. (b) Filtering and noise removal eliminate the localized noise. (c) and (d) show the plasma line detection using the unsupervised methodology illustrated in Figure 2.

(IQR) is the distance between these quartiles. Outliers, defined as samples exceeding $1.5 \cdot$ IQR from the top or bottom of the box, are represented by the symbol 'o'. Additionally, the whiskers extend from each box: one connects the upper quartile to the non-outlier maximum (the highest value that is not an outlier), and the other connects the lower quartile to the non-outlier minimum (the lowest value that is not an outlier).

As seen in Figure 10, plasma line intensities are grouped into three altitude ranges. $66\,\%$ of the plasma lines are detected
between $233.1–257.8\,\mathrm{km}$ (corresponding to the range around the F2 peak), $22\,\%$ detected between $208.5–233.2\,\mathrm{km}$ (below the F2 peak) and $12\,\%$ detected between $257.8–282.5\,\mathrm{km}$ (above the F2 peak). As can be seen in Figure 4, the intensity of the plasma line is largest around the F2 density peak, which explains why most of the plasma lines are observed in the range $233.1–257.8\,\mathrm{km}$. As the resonance phase energy increases, the plasma line intensity increases for all the altitude groups, peaking around a phase energy of $3\,\mathrm{eV}$, after which it starts decreasing again. This result is consistent with the model predic-
tions by Nilsson et al. (1996) where F-region photoelectron-enhanced plasma lines observed by EISCAT UHF radar peaks at approximately $2.7\,\mathrm{eV}$. The variation in plasma line intensity can be attributed to the effect of Landau damping, where electro-



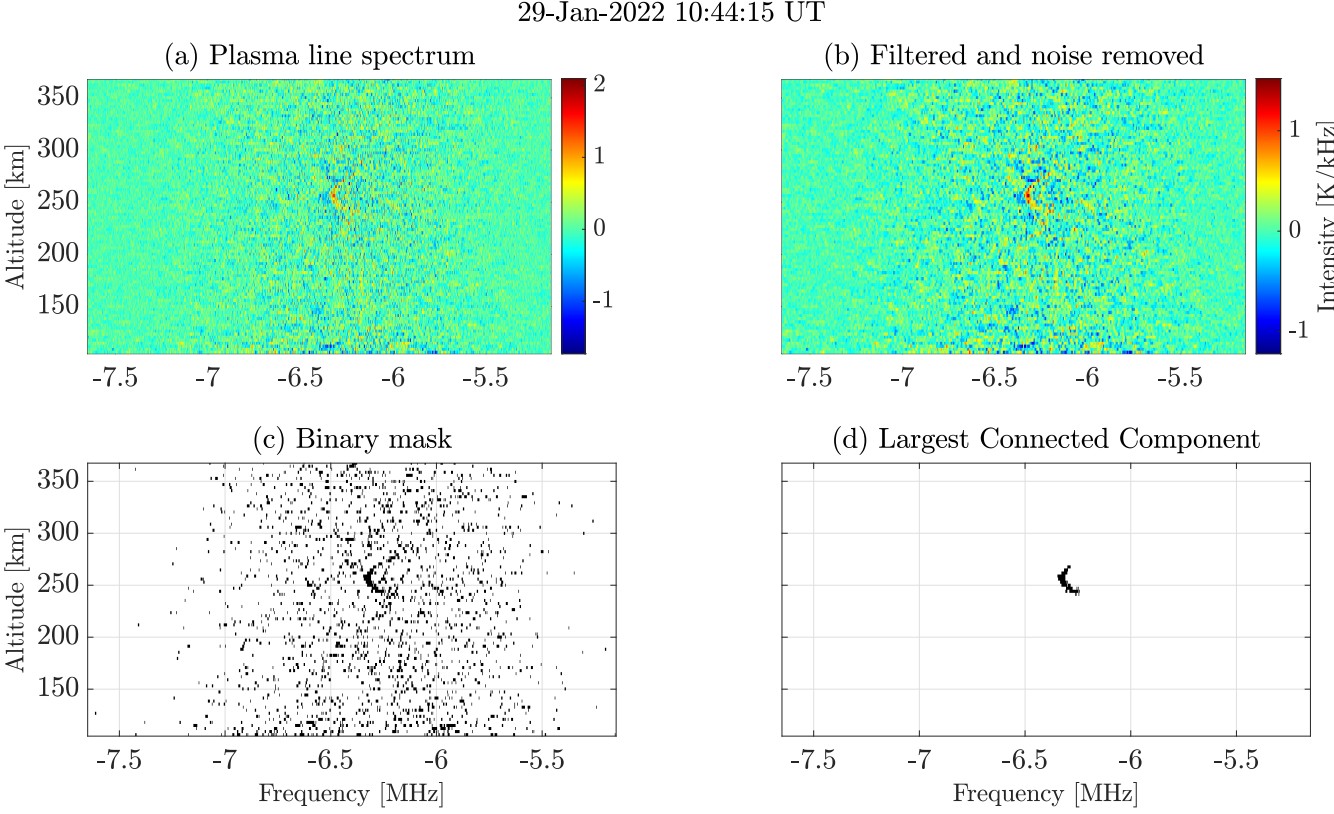

**Figure 9.** (a) Plasma line spectrum from EISCAT UHF radar with high noise around the centre frequency from $-7$ to $-6$ MHz. (b) The filtering and noise removal method effectively enhances the plasma line while suppressing the noise. (c) and (d) show the plasma line detection using the unsupervised methodology illustrated in Figure 2.

static wave energy is absorbed by resonant electrons. The magnitude of Landau damping depends on the slope of the reduced one-dimensional velocity distribution function of the electrons along the direction of the scattering wave vector. Smaller Landau damping corresponds to larger plasma line intensity. The presence of a suprathermal electron population introduces a
high-energy tail to the distribution function, decreasing the slope at higher energies and thereby, reducing Landau damping. Consequently, as phase energy increases, the plasma line intensity increases. It is also worth noting there do not seem to be any significant or systematic differences in plasma line intensity between the altitude groups for any given energy, indicating that the suprathermal populations are very similar at these altitudes.

Figure 11 presents plasma line intensity as a function of phase energy for the available aspect angles and integrated over
altitude and the six days from 26-31 January 2022 07-15 UT. The data is visualized as box plots similarly to Figure 10. It can be seen that the plasma line intensity generally decreases as the aspect angle increases. This result is consistent with the findings of Yngvesson and Perkins (1968); Fredriksen et al. (1992). Yngvesson and Perkins (1968) suggested that the Landau damping increases with the aspect angle. This implies that the reduced one-dimensional electron velocity distribution function becomes







**Figure 10.** Plasma line intensity $k_B T_p$ as a function of phase energy $E_\phi$, integrated over time and aspect angles, and binned by altitude.







**Figure 11.** Plasma line intensity $k_B T_p$ as a function of phase energy $E_\phi$, integrated over time and altitude, and binned by aspect angle.





steeper at higher aspect angles, meaning that more electrons absorb energy from the wave. This can be a possible reason for
the reduction in plasma line intensity at higher aspect angles.

Even though the data is taken from multiple days covering different phase energy intervals, there is an overall smooth
variation of plasma line intensity with phase energy as seen in Figures 10 and 11. This suggests that the suprathermal electron
velocity distribution at the corresponding velocities does not vary significantly over the observation campaign.

Previous studies (Nilsson et al., 1996; Guio et al., 1998; Guio and Lilensten, 1999) have modelled plasma line intensity
using suprathermal electron population angular differential energy fluxes computed by an ionospheric electron transport code
(Lummerzheim and Lilensten, 1994). Building on the plasma line model described in Guio et al. (1998), we aim to further
develop an aspect angle dependent plasma line intensity model and apply it to the current dataset. This approach, which will
be detailed in a future paper, will use the general-purpose plasma line analysis pipeline we developed here.

The methodologies presented here have facilitated the detection of hundreds of plasma line events, significantly increasing
the volume of data available for theoretical validation compared to previous studies. The only limitation of the dataset is the
number of pointing directions to the magnetic field which is imposed by the experiment design. This is where advanced radar
systems like EISCAT 3D (McCrea et al., 2015) will be useful. Unlike the current EISCAT systems with mechanically steered
antennas, EISCAT 3D with phased array antenna technology will enable the beam to be steered at different aspect angles with
higher time resolution. This capability will greatly expand the dataset for studying the aspect angle dependence of plasma line
parameters.

*Code and data availability.* The GUISDAP software, used to obtain the ISR spectra, is available for download at https://sourceforge.net/
projects/guisdap/. The EISCAT data used here are publicly available and can be accessed from the EISCAT data portal at https://portal.eiscat.
se/schedule/ under archived data for UHF radar. The data and software used in this paper (Gupta, 2024) are available for reviewers at the
following link: https://doi.org/10.5281/zenodo.14135585.

*Author contributions.* PG and MG conceptualized the project. MG was responsible for data curation, formal analysis, investigation, method-
ology development, software creation, validation, and visualization. PG provided project administration, resources, and supervision. MG
prepared the original draft of the manuscript, and both PG and MG reviewed and edited the final manuscript.

*Competing interests.* The authors declare that they have no conflict of interest.

*Acknowledgements.* The authors thank Dr Ingemar Häggström for his guidance in analyzing the EISCAT radars plasma line data. The authors
also thank Dr Juha Vierinen for his feedback on the manuscript.



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
