# Peer review of "Data reduction of incoherent scatter plasma line parameters"

_EGUsphere, 2025_

## Referee Comment (RC1)

**Review of "Data reduction of incoherent scatter plasma line parameters" by Gupta and Guio**

Summary: This paper presents new methods for detecting and analyzing plasma lines with the EISCAT radar. One method involves supervision and hand tuning of parameters but accurately extracts any plasma lines present. The second method is fully automated and can be applied unsupervised to a large data set, but suffers from failed detections. The authors apply these new algorithms to 5 days of plasma lines observed by the EISCAT UHF radar. The extracted plasma line frequency and intensity are then analyzed, showing the dependence of the radar pointing direction and ionospheric density.

This is a well written paper that showcases a new method for plasma line extraction and further shows how plasma lines are influenced by radar and ionospheric parameters. I recommend this paper for publication with some revisions.

Review Comments:

1. In Section 3.3, it is mentioned that the plasma lines are fit to Gaussian functions. Can the authors comment on why a Gaussian is chosen instead of a Lorentzian? From *Perkins and Salpeter* (1965), the frequency spectrum of a plasma line is expected to be a Lorentzian shape.

2. Figure 4 shows the fitted plasma lines, but this is hard to see as the red curves are thick and cover most of the data. This figure would be more clear if modified so that both curves are visible. Suggestions are to plot the red curve underneath the black curve, reduce the thickness of the red curve, or make the red curve a dashed line.

3. In Section 3.4, the phase energy is given by equation 16. First, it is not necessary to approximate the Bragg wavelength as $2f_{radar} + f_{offset} \rightarrow 2f_{radar}$. Second, as shown in *Longley et al.* 2021, the phase energy has a strong dependence on aspect angle. This dependence is given in equation 2 of *Longley et al.* 2021 (ignoring the gyro terms):

$$E_\phi = \frac{1}{2} m_e \left( \frac{\omega}{k cos^2 \theta} \right)^2$$

It is recommended the authors correct equation 16 for both of these changes, while also commenting on the aspect angle changing what electron energies are resonant with the plasma line. Note that this effect is small for the experiments in this paper, but may change how Figures 10 and 11 look.

4. In line 235, it is mentioned that plasma lines are only detected in the 1.7-3.4 eV range. Can the authors comment of where the ionospheric densities were high enough to observe plasma lines at the upper limit of the filter band (7.65 MHz)? Also, it should be noted that the plasma lines with phase energies below 1.7 eV correspond to a lower density plasma, and therefore lower SNR.

5. The interpretation of plasma line enhancement given in lines 286-293 is not consistent with the derivations of *Longley et al.* 2021. There is a distinction between the total damping of Langmuir waves and the individual contributions: collisional

damping, thermal Landau damping, and photoelectron Landau damping. Plasma lines are enhanced above thermal levels because the photoelectron Landau damping term flips signs at photoelectron peaks and generates waves through inverse Landau damping. It is suggested that the authors revise this paragraph.

6. A similar issue of interpretation comes in the paragraph around line 295. Yngvesson and Perkins 1968 assumed suprathermal electrons would be unmagnetized, and therefore their claims about changes with aspect angle are not valid. This was corrected with a fully magnetized derivation in *Longley et al.* 2021, which has some plots showing that plasma line intensity is roughly constant from 0 to 10 degrees aspect angle, then increases from 10 to 20 degrees. However, the full understanding of the presented EISCAT UHF observations is difficult as a good photoelectron model is needed for plasma line power calculations. The authors should revise the discussion in this paragraph to avoid repeating the incorrect conclusions of Yngvesson and Perkins.

7. Lines 306-307 state "we aim to further develop an aspect angle dependent plasma line intensity model and apply it to the current dataset." Can the authors clarify how such a model would be different from either *Enger* 2020 (https://hdl.handle.net/10037/19542) or *Longley et al.* 2021?

---

## Author Comment (AC1)

**Response to the reviewers**

The authors would like to thank the reviewer for constructive comments and suggestions that have helped improve the quality of this manuscript which will be revised accordingly. Please see below our responses. Reviewer comments are reproduced, and our responses are given in blue below each comment.
* * *
**Reviewer 1**

**Summary:** This paper presents new methods for detecting and analyzing plasma lines with the EISCAT radar. One method involves supervision and hand tuning of parameters but accurately extracts any plasma lines present. The second method is fully automated and can be applied unsupervised to a large data set, but suffers from failed detections. The authors apply these new algorithms to 5 days of plasma lines observed by the EISCAT UHF radar. The extracted plasma line frequency and intensity are then analyzed, showing the dependence of the radar pointing direction and ionospheric density.

This is a well written paper that showcases a new method for plasma line extraction and further shows how plasma lines are influenced by radar and ionospheric parameters. I recommend this paper for publication with some revisions.

**Reviewer Comment 1:** — In Section 3.3, it is mentioned that the plasma lines are fit to Gaussian functions. Can the authors comment on why a Gaussian is chosen instead of a Lorentzian? From Perkins and Salpeter (1965), the frequency spectrum of a plasma line is expected to be a Lorentzian shape.

**Reply:** While Lorentzian functions are theoretically expected for plasma line spectra (Perkins and Salpeter, 1965), we think Gaussian functions are easier for interpretation since they provide parameters such as plasma line frequency, width, and intensity in a straightforward manner. In addition, as shown in Figure 4, there is quite a lot of noise as soon as the Gaussian fits fall to zero. Because Gaussian functions decay more rapidly to zero than Lorentzians, they are less sensitive to this noise and therefore do not overestimate intensity. Furthermore, the observed spectral width is not the natural width of a plasma line at a specified frequency and height, but rather a smeared version resulting from frequency variations across the radar range gate and the finite integration time. For these reasons, we consider Gaussian fits to be the more reliable choice for the present analysis, a choice also adopted by Guio et al. (1996).

**Reviewer Comment 2:** — Figure 4 shows the fitted plasma lines, but this is hard to see as the red curves are thick and cover most of the data. This figure would be more clear if modified so that both curves are visible. Suggestions are to plot the red curve underneath the black curve, reduce the thickness of the red curve, or make the red curve a dashed line.

**Reply:** In the revised manuscript, we will improve Figure 4 by placing the red curve beneath the black curve and reducing its thickness. This will make both the experimental data and the fitted plasma lines more clearly visible.

**Reviewer Comment 3:** — In Section 3.4, the phase energy is given by equation 16. First, it is not necessary to approximate the Bragg wavelength as $2f_{\text{radar}} + f_{\text{offset}} \rightarrow 2f_{\text{radar}}$. Second, as shown in Longley

et al. 2021, the phase energy has a strong dependence on aspect angle. This dependence is given in equation 2 of Longley et al. 2021 (ignoring the gyro terms):

$$E_\phi = \frac{1}{2} m_e \left( \frac{\omega}{k \cos^2 \theta} \right)^2 \tag{1}$$

It is recommended the authors correct equation 16 for both of these changes, while also commenting on the aspect angle changing what electron energies are resonant with the plasma line. Note that this effect is small for the experiments in this paper, but may change how Figures 10 and 11 look.

**Reply**: We agree that the approximation of the Bragg wavelength as $2f_{radar} + f_{offset} \to 2f_{radar}$ is not strictly necessary. Nevertheless retaining the offset frequency $f_{offset}$ of the receiver with respect to the transmitted frequency $f_{radar}$ is important to understand the physics of the scattering process where the scattering wavevector is

$$\vec{k_s} = \vec{k_r} - \vec{k_t}. \tag{2}$$

The received and transmitted wavenumbers are $k_r = 2\pi(f_{radar} + f_{offset})/c$ and $k_t = 2\pi f_{radar}/c$ respectively. For backscattering the wavevectors are colinear and the scattering wavenumber is $k_s = k_r + k_t = 2\pi(2f_{radar} + f_{offset})/c$ which can be written $k_s = 2k_r + \Delta k$ with the fractional correction

$$\frac{\Delta k}{k_r} = \frac{f_{offset}}{f_{radar}}. \tag{3}$$

Since in our experiment $f_{radar} = 931$ MHz and $f_{offset}$ is only a few MHz, the correction introduced by the frequency offset at which plasma line is observed is of the order or less than $1\%$.

We will clarify this point to make explicit that, while preferable to retain the full Bragg expression for lower-frequency radars, such as VHF, its impact is negligible for this study with the UHF radar.

Regarding the aspect-angle dependence, we note that the expression quoted in the reviewer's comment contains a $\cos^2 \theta$ term in the denominator. We believe this may have been a typographical error and interpret the intended form as

$$E_\phi = \frac{1}{2} m_e \left( \frac{\omega}{k \cos \theta} \right)^2. \tag{4}$$

We believe this expression refers to only the parallel component of phase energy. In our analysis, however, we consider as the main parameter the total phase energy, defined as

$$E_\phi = \tfrac{1}{2} m_e v_\phi^2 = \frac{1}{2} m_e \left( \frac{\omega}{k_s} \right)^2, \tag{5}$$

where $k_s$ is the scattering wavenumber defined above. The expression for the phase velocity $v_\phi$ is the same as specified in Eq. (4) of Akbari et al. (2017). The phase velocity of the scattered wave is defined in Eqs. (14)—(15) in our manuscript. The data are then binned with respect to both total kinetic energy and aspect angle, so the aspect-angle dependence is naturally accounted for in the results. Therefore, we consider it most appropriate to retain the total phase energy and not multiply the wavenumber by $\cos \theta$.

**Reviewer Comment 4:** — In line 235, it is mentioned that plasma lines are only detected in the 1.7–3.4 eV range. Can the authors comment of where the ionospheric densities were high enough to observe plasma

lines at the upper limit of the filter band (7.65 MHz)? Also, it should be noted that the plasma lines with phase energies below 1.7 eV correspond to a lower density plasma, and therefore lower SNR.

**Reply**: The highest-frequency plasma line observed in our dataset occurred at 6.7598 MHz (line 224) at an altitude of 245.6 km, which lies near the middle of the altitude range where plasma lines were detected. Plasma frequencies above 6.74 MHz were observed between approximately 245.6 and 260.3 km.

We agree that plasma lines with phase energies below 1.7 eV may be present but obscured by noise. In principle, plasma lines from the E-region should also be observable, but in our measurements, they were not detected. This might be because for the relatively low electron densities in the E-region, the corresponding plasma line frequencies could fall outside the receiver filter band (below 2.75 MHz). Also, the E-region is characterised by larger electron-neutral collision frequency which can weaken the suprathermal enhancement mechanism, so that any signal would remain below the noise floor. These might be the reasons that all plasma lines detected in this study were confined to the F-region.

**Reviewer Comment 5:** — The interpretation of plasma line enhancement given in lines 286–293 is not consistent with the derivations of Longley et al. 2021. There is a distinction between the total damping of Langmuir waves and the individual contributions: collisional damping, thermal Landau damping, and photoelectron Landau damping. Plasma lines are enhanced above thermal levels because the photoelectron Landau damping term flips signs at photoelectron peaks and generates waves through inverse Landau damping. It is suggested that the authors revise this paragraph.

**Reply**: We agree that plasma line enhancement involves several distinct damping and excitation mechanisms, namely collisional damping, thermal Landau damping, and photoelectron Landau damping, as outlined in Longley et al. (2021).

In a 1D reduced velocity distribution obtained by integrating an approximately isotropic 3D electron VDF, the classical inverse Landau damping mechanism described by Longley et al. (2021) does not strictly occur, as fine structures in the velocity distribution (peaks and valleys) are largely averaged out during the integration.

However, we would like to emphasise that the phase energies investigated in our study (less than 4 eV) are lower than those discussed in Longley et al. (2021). In this regime, the dominant effects are Landau damping and Cerenkov excitation (Nilsson et al., 1996), rather than variations in the suprathermal tail. As summarised by Akbari et al. (2017):
*"variation of $T_p$ with phase velocity generally abides by the following trend: at the lowest phase velocities (i.e., highest probing wavenumbers, assuming a fixed electron density) thermal electrons and their Cerenkov excitation and Landau damping are the dominant terms and $T_p$ is close to its thermal level $T_e$. As the phase velocity increases, the suprathermal electrons become increasingly important and $T_p$ increases as the excitation and damping by suprathermal electrons dominate the numerator and the denominator, respectively. At sufficiently large phase velocities, where the number of resonant electrons significantly drops, on the other hand, the collisional terms dominate both the excitation and damping and the plasma line temperature approaches $T_e$ once again."*

Given our relatively high probing wavenumbers, the observed enhancements are most consistently explained by Landau damping and Cerenkov excitation at low phase energies (lifting of the tail of electron VDF), rather than by inverse Landau damping of photoelectron peaks. We will revise lines 286–293 to clarify this interpretation.

**Reviewer Comment 6:** — A similar issue of interpretation comes in the paragraph around line 295. Yngvesson and Perkins 1968 assumed suprathermal electrons would be unmagnetized, and therefore their claims about changes with aspect angle are not valid. This was corrected with a fully magnetized derivation in Longley et al. 2021, which has some plots showing that plasma line intensity is roughly constant from 0 to 10 degrees aspect angle, then increases from 10 to 20 degrees. However, the full understanding of the presented EISCAT UHF observations is difficult as a good photoelectron model is needed for plasma line power calculations. The authors should revise the discussion in this paragraph to avoid repeating the incorrect conclusions of Yngvesson and Perkins.

**Reply**: We would like to clarify that the analysis in Yngvesson and Perkins (1968) does account for the magnetisation of photoelectrons, as reflected in their Eq. 20. Therefore, their treatment of aspect-angle dependence is valid and should not be considered incorrect.

In our study, for the phase energies and probing wavenumber considered (less than 4 eV), the dominant mechanisms explaining the observed plasma line enhancements are Landau damping and Cerenkov excitation, as highlighted in the previous comment.

**Reviewer Comment 7:** — Lines 306-307 state "we aim to further develop an aspect angle dependent plasma line intensity model and apply it to the current dataset." Can the authors clarify how such a model would be different from either Enger 2020 (https://hdl.handle.net/10037/19542) or Longley et al. 2021?

**Reply**: The key difference in our approach compared to Longley et al. (2021) lies in the treatment of the photoelectron distribution. Longley et al. used a distribution derived from AE-C and AE-E satellite measurements at a single altitude and solar zenith angle from a time (Figure 1 adapted from Hernandez, 1983) years before the plasma line observations (March 17, 2015). Moreover, in their figure, the distribution does not appear to include pitch-angle dependence.

In the MSc thesis of Enger (2020), the electron transport model code Aurora was used, but pitch-angle variation in the electron VDF was not accounted for, with only the component along the magnetic field considered. Both studies, therefore, effectively assume isotropy in their calculations.

In contrast, our model is a direct extension to directions oblique to the magnetic field of the framework published in Guio et al., 1998 and Guio and Lilensten, 1999 which treated the field-aligned direction only. Our model uses the latest version of the multistream numerical electron transport model that was used by Guio, 1998 and Guio, 1999 based on Lummerzheim et al., 1994. The model provides the electron VDF as a function of altitude, energy, and a discrete set of pitch angles spanning 0–180°, taking into account local conditions, Ap, and F10.7 indices. This allows us to capture fine-scale variations in the electron distribution that are relevant to the observed plasma line intensities.

Additionally, we will implement a dynamic calculation of the summation limits of the scaled Bessel functions used in the theoretical incoherent scatter spectral modelling. Unlike Longley et al., who used fixed limits ($\ell = \pm 30$), our approach dynamically calculates the limits based on the electron VDF and the wavenumber and aspect angle, ensuring that all relevant features of the electron distribution are included in the calculation, while avoiding unnecessary computations when the distribution becomes negligible.

Overall, our model will extend previous work by incorporating both anisotropic electron distributions and a more flexible computational framework.

---

## Author Comment (AC2)

**Response to the reviewers**

The authors would like to thank the reviewers for constructive comments and suggestions that have helped improve the quality of this manuscript which will be revised accordingly. Please see below our responses. Reviewer comments are reproduced, and our responses are given in blue below each comment.
* * *
**Reviewer 2**

The manuscript "Data reduction of incoherent scatter plasma line parameters" presents a study of plasma line detection in the EISCAT UHF radar dataset, using two approaches: a supervised (kind of manually adjusted) and an unsupervised technique (with the detection based on the statistical parameters of the data itself).

The manuscript is well written and easy to follow. The figures are generally clear and of good quality. The conclusions are well supported by the data. Minor revisions/clarifications could improve the quality of the manuscript, which is of high standard already.

**Reviewer Comment 1:** — As the paper provides an important reference for plasma line analysis, it would be good to mention in the introduction plasma lines associated with instabilities and turbulence. The interested reader will find this information in e.g. the review by Akbari et al., but it would be good to mention this fact – and maybe even add references to key publications.

**Reply**: We will add a short note in the introduction at line 42:

*"Plasma lines may also be strongly enhanced by plasma instabilities and Langmuir turbulence. Such processes can lead to intensities several orders of magnitude above the thermal level, and have been studied both theoretically and observationally. For example, Guio and Forme (2006) presented a numerical study of Langmuir turbulence driven by low-energy electron beams, while Isham et al. (2012) reported the first direct evidence of naturally occurring cavitating Langmuir turbulence in the ionosphere. A comprehensive review of these processes is given by Akbari et al. (2017)."*

We will include these references in the revised manuscript to provide readers with a starting point to further study.

**Reviewer Comment 2:** — The discussion of Figure 7 is not very clear in lines 224–227. The description mentions an increase of the plasma line frequency at "about 11:20:15 UT", and then mentions a "gradual increase" from 10:00-11:20 UT. Both are related to the Sun, one to the "solar EUV radiation" and the other to "sun exposure". In my opinion, this description should be consolidated.

**Reply**: In the revised manuscript, we will consolidate the description in lines 224–227 to read:
*"The plasma line frequency increases gradually from 10:00-11:20 UT (Figure 7), driven by an increase in the photoelectron population from solar EUV radiation."*

**Reviewer Comment 3:** — An interesting phenomenon is observed in Figure 7, namely the undulation of the location of detected plasma line. This would indicate that the peak of the profile is moving up and down in altitude. Some quasiperiodic variations of the lower "boundary" of the F region are seen also in the top panel of Figure 6. While not directly related to main subject of the discussion, the authors may want to

comment on possible nature of the variations (could these be related to gravity waves? or variations in the ionospheric convection? Or yet something else?). Are these variations of the electron concentration profile capable of affecting the plasma line enhancement condition?

**Reply**: We thank the reviewer for highlighting the undulations in the location of the detected plasma line. We agree that this is an interesting feature and note that similar variations were observed on other days of the experiment as well. We are currently uncertain about the exact cause of the observed undulations. If the reviewer is aware of any studies reporting similar variations in background electron density, we would be grateful for references and happy to consider including them. To our knowledge, we are not aware of prior studies documenting this phenomenon, and we are currently investigating it further with colleagues.

The plasma line enhancement condition primarily depends on the presence of suprathermal electrons and the shape of their velocity distribution function (VDF). Variations in the F-region lower boundary or undulations in the electron density profile do not alter this condition.

As discussed in the manuscript, processes such as solar EUV radiation and auroral precipitation generate suprathermal electrons, which lead to observable plasma line enhancements. Therefore, while processes that cause altitude undulations (e.g., gravity waves or ionospheric convection) may affect the local electron density profile, they do not change the fundamental requirement for plasma line enhancement: the presence of suprathermal electrons. The enhancement mechanism itself remains governed by the suprathermal electron VDF.

**Reviewer Comment 4:** — A note on Fig. 6 is that the date of the production of the figure is not important, and probably should not be included in the plot.

**Reply**: The production date will be removed from Figure 6 in the revised manuscript.

**Reviewer Comment 5:** — For Figure 7 it is a good idea to have a better setup of the time labels, having only two of them, at 10:29 and 11:30 does not help in pointing out specific times in the plot.

**Reply**: In the revised manuscript, we will adjust the time axis of Figure 7 by adding major ticks and labels every 30 minutes and minor ticks every 5 minutes, making it easier to identify specific times in the plot.

**Reviewer Comment 6:** — The bottom panel indicates the intensity units in eV, is this correct? Figure 4 gives intensity in K/kHz, while there is also an extensive discussion of the phase energy" (eq. 16). In Figure 5 the intensity is in K, and Figures 10 and 11 present $k_B T_p$.

**Reply**: The units in Figure 7 are correct. Figure 4 shows the plasma line antenna temperature, $T_A^p$, in units of K/kHz. Integrating this over frequency produces the altitude profiles in units of K shown in Figure 5. These profiles are then converted to plasma line temperature, $k_B T_p$, expressed in eV, which is displayed in Figure 7. We acknowledge that the caption of Figure 7 incorrectly referred to the intensity as $A_p$. The plotted quantity is actually the plasma line intensity $k_B T_p$, i.e. the plasma line temperature in energy units (eV) derived from $A_p$ expressed in antenna temperature $T_A^p$ and using Eqs. (12)–(13). This will be corrected in the revised manuscript.

$k_B T_p$ is derived from the amplitude of the plasma lines, whereas the phase energy is obtained from the resonance frequency $f_r$ (Figures 4 and 5) using Eqs. (14)–(16). We will harmonise the notation throughout the manuscript to improve consistency.

**Reviewer Comment 7:** — A minor detail: text predominantly refers to "altitudes", while some of the figures are marked with "height" (while others with "altitude"). This could probably be harmonized.

**Reply**: We will harmonise the terminology throughout the manuscript and figures, consistently using "altitude" to improve clarity.

**Reviewer Comment 8:** — The paragraphs in lines 228–236 seem somewhat misplaced. In part they refer to "similar plots" which are not presented, in contrast to the detailed discussion of the even presented.

**Reply**: The reason for not including plots from all six days in the manuscript was to avoid excessive length, as these additional figures do not provide new information beyond what is already shown. In the dataset and model code, the figures for the other days can already be generated by modifying the argument of the `fitted_params` function in `paper1.m`. We will make these figures available as supplementary material for the reviewer's reference if desired.

**Reviewer Comment 9:** — They also refer to Figure 10, not to be presented to the reader until later in paper. The consequences of observing a narrower interval of the phase energy range than covered by the system is not very clear at this point. A suggestion would be moving this material to the discussion (line 257 and onwards), rather than having it in Results.

**Reply**: We will move the lines 233–236 and consolidate them with the discussion of Figure 10, starting from line 279. The intention of mentioning the limited phase energy range in the Results section was to highlight that, even with multiple days of observations, the daytime ionospheric conditions between consecutive days were relatively similar. To explore a wider range of phase energies, it would be necessary to have observations covering a broader range of ionospheric conditions.

**Reviewer Comment 10:** — I suggest considering renaming the last section to something like "Summary and Discussion". Conclusions would be expected to be a concise summary of the findings, the nature of the presentation here is more of a discussion (among other aspects, four new figures are presented!).

**Reply**: We agree and will rename the last section to "Summary and Discussion" in the revised manuscript.

**Reviewer Comment 11:** — The purpose of showing presenting Figures 8 and 9 in the concluding section is not very clear (paragraph in lines 254–256). Probably they would fit better in the analysis section?

**Reply**: We agree that Figures 8 and 9 would fit better in the Results section. In the revised manuscript, we will move these figures. These figures intend to illustrate the robustness of the methods against some common data analysis challenges encountered in plasma line detection.

**Reviewer Comment 12:** — Line 262–263 "The absence of upshifted plasma lines means that we cannot use our data to calculate ionospheric currents." – what this ever the intension in this work?

**Reply**: Initially, the authors intended to calculate ionospheric currents using plasma lines observed at both upshifted and downshifted frequencies. Our methodology is designed to work when plasma lines are detected at either frequency shifts. However, as explained in lines 259-–263, this was not possible due to the absence of upshifted plasma lines in our observations.

**Reviewer Comment 13:** — Figures 10 and 11 present $k_B T_p$ in eV. These would correspond to temperatures well in excess of those shown in Fig. 5 (left panel). I suspect that the numbers are not directly comparable, but it would be helpful with a clarification in the paper on how they should relate to each other.

**Reply**: The temperatures shown in Figure 5 correspond to the plasma line antenna temperature $T_A^p$, whereas Figures 10 and 11 present the plasma line temperature $k_B T_p$. These quantities are related via equations (12) and (13), and Section 3.4 discusses how they are connected. In the revised manuscript, we will clarify this distinction in the figure captions to make it explicit which temperature each figure represents, improving clarity for the reader.

**Reviewer Comment 14:** — It would be interesting to discuss how the choice of the dataset affects the results. The observations are from several days in a row in January, all during daytime. How would the observations look in a different seasons and/or time of the day?

**Reply**: In the revised manuscript, we will add this in the "Summary and Discussion" section:

*"Plasma line enhancements above the thermal level depend directly on the suprathermal electron flux. During daytime, the prolonged input of solar EUV energy produces suprathermal electron fluxes over an extended period, resulting in more frequent plasma line detections, particularly around the F-layer peak where electron density is highest. At night, suprathermal electron flux is generated through transient auroral activity, resulting in observable plasma line enhancements. For nighttime observations, careful selection of the integration time is necessary, as too long an interval could average out short-lived enhancements and make plasma lines harder to detect.*

*Regarding seasonal variations, we expect statistical plasma line occurrence to vary in line with the study from Ivchenko et al. 2018 (Figure 2). In particular, May–July will offer the highest number of plasma line observations due to prolonged solar illumination and high suprathermal electron fluxes. Both solar EUV radiation and auroral precipitation can contribute during March–April and August–October, allowing plasma line detection under a wider range of conditions. Therefore, there can be more observations over wider phase energy intervals to build figures similar to Figures 10 and 11 for plasma lines observed due to both solar EUV radiation and auroral precipitation.*

*The primary objective of the present paper is to develop and demonstrate methodology for plasma line detection and analysis. The methodology presented can be applied in future long-term studies to investigate how plasma line occurrence and phase energy distributions vary with season, time of day, and aspect angle. A proper exploration of the seasonal and diurnal variations would require multiple observations spanning the year under varying conditions, which is beyond the scope of the current work but could be pursued in future studies."*